



**Identifying and Quantifying Source Contributions of Air Quality Contaminants**
**during Unconventional Shale Gas Extraction**
Nur H. Orak[1,*], Matthew Reeder[2,3], Natalie J. Pekney[3]
[1] [*]Corresponding author: Tel: +90-216-4140545, nho@alumni.cmu.edu
Marmara University, Department of Environmental Engineering
[2] Leidos Research Support Team, National Energy Technology Laboratory,
Pittsburgh, PA
[3]U.S. Dept. of Energy National Energy Technology Laboratory, Pittsburgh, PA



**Abstract**
The United States experienced a sharp increase in unconventional natural gas (UNG)
development due to the technological development of hydraulic fracturing
("fracking"). The objective of this study is to investigate the effect of unconventional
natural gas development activities on local air quality as observed at an active
Marcellus Shale well pad at the Marcellus Shale Energy and Environment Laboratory
(MSEEL) in Morgantown, Western Virginia, USA. Using an ambient air monitoring
laboratory, continuous sampling started in September 2015 during horizontal drilling
and ended in February 2016 when wells were in production.  High resolution data
were collected for the following air quality contaminants: volatile organic compounds
(VOCs), ozone ($O_3$), methane ($CH_4$), nitrogen oxides (NO and $NO_2$), carbon dioxide,
($CO_2$), as well as typical meteorological parameters (wind speed/direction,
temperature, relative humidity, and barometric pressure). Positive Matrix
Factorization (PMF), a multivariate factor analysis tool, was used to identify possible
sources of these pollutants (factor profiles) and determine the contribution of those
sources to the air quality at the site. The results of the PMF analysis for well pad
development phases indicate that there are three potential factor profiles impacting air
quality at the site: *natural gas*, *regional transport/photochemistry*, and *engine*
*emissions*. There is a significant contribution of pollutants during horizontal drilling
stage to *natural gas* factor. The model outcomes show that there is an increasing
contribution to *engine emission* factor over different well pad drilling through
production phases. Moreover, model results suggest that the major contributions to
the *regional transport/photochemistry* factor occurred during horizontal drilling and
drillout stages.
Keywords: ambient monitoring; natural gas; air pollution; source apportionment





**Introduction**

There is a rapid increase in unconventional natural gas exploration by recent

technological advances (USEIA 2020). The success of the US in exploiting

unconventional natural gas has stimulated other countries. As a result, there is a

growing attention by public for the potential public health impacts of UNG extraction.

In response to emerging public concern regarding the process of fracking for UNG

extraction, several studies have investigated the potential public health risks of UNG

development (Adgate et al. 2014; Hays et al. 2015; Hays et al. 2017; Werner et al.

2015). A part of adverse health effects are related to exposure of environmental

pollution (Elliott et al. 2017; Elsner and Hoelzer 2016; Paulik et al. 2016). The

majority of environmental impact studies focus on water quality impacts of

unconventional natural gas development (Annevelink et al. 2016; Butkovskyi et al.

2017; Jackson et al. 2015; Torres et al. 2016). However, relatively fewer studies focus

on air quality impacts (Islam et al. 2016; Ren et al. 2019; Swarthout et al. 2015;

Williams et al. 2018). Some studies focus on collecting and analyzing data for pre-

operational phase of fields to provide baseline dataset for future work that operational

shale gas activities can be later evaluated  (Purvis et al. 2019). Non-methane

hydrocarbons (NMHC) and nitrogen oxides ($NO_x$) are of most interest as some

NMHC can be toxic (such as benzene) (P. M. Edwards et al. 2014), therefore, several

studies focuses on increases in methane, NHMC, and ozone in oil and gas producing

regions (Pacsi et al. 2015; Roest and Schade 2017). Another study explored the

importance of the deployment autonomy of portable measurement systems by

measuring exposure upwind, within and downwind of operation of hydraulic

fracturing equipment to protect workers (Ezani et al. 2018). There are also more

comprehensive studies for data collection. Swarthout et al. (2015) conducted a field

campaign to investigate the impact of UNG production operations on regional air

quality. Highest density of methane, carbon dioxide, and volatile organic carbons



(VOCs) were observed closer to UNG wells. A limited number of studies available on
source apportionment for major air pollutants (Majid et al. 2017; Prenni et al. 2016).
These studies have lacked a comparison of the effects during distinct operational
phases of natural gas extraction: well pad construction, drilling (vertical and
horizontal), well stimulation (hydraulic fracturing followed by flowback), and
production.
Several compounds are associated with emissions from each phase of well installation
and development, depending on the activity and equipment in use for each phase.
Activities that require the use of off-road diesel construction vehicles have emissions
of coarse particulate matter ($PM_{10}$ aerodynamic diameter $\leq 10$ μm) from the
suspension of dust from vehicle traffic on dirt and gravel roads, as well as volatile
organic compounds (VOCs), nitrogen oxides ($NO_x$) and fine particulate matter
smaller than 2.5 μm in aerodynamic diameter ($PM_{2.5}$) from the vehicle exhaust.
During vertical and horizontal drilling, there are emissions of $NO_x$, $PM_{2.5}$, and VOCs
from diesel powered drilling rigs, and fugitive emissions of natural gas (methane
($CH_4$) and other hydrocarbons). Hydraulic fracturing activities add emissions from
truck traffic and diesel-powered compressors ($NO_x$, $PM_{10}$, $PM_{2.5}$, VOCs).  Emissions
of VOCs and $CH_4$ from water separation tanks, venting, and degassing of produced
waters occur during flowback operations.  In addition to these primary sources of
emissions at the site, secondary production of ozone ($O_3$) and $PM_{2.5}$ from
photochemistry can result from emissions during any of the operational phases.
This is the first study, to our knowledge, to collect high time resolution ambient
concentrations of compounds emitted from well pad activity during various phases of
operation such that the relative air quality effect of each phase of development can be
investigated. This detailed information about the distribution of emission sources'





impact through a well pad's development phases is needed to manage the associated
risks from emissions.
**Methods**
**Monitoring Location: Marcellus Shale Energy and Environment Laboratory**
The Marcellus Shale formation covers an area of approximately 240,000 km$^2$ across
several states: New York, Pennsylvania, Ohio, West Virginia, Maryland, and
Virginia (Kargbo et al. 2010)(Figure S1). The Marcellus Shale Energy and
Environment Laboratory (MSEEL) is an approximately 14,000 m$^2$ study well pad in
Morgantown, WV, USA (39.602$^0$ N, 79.976$^0$ W) (MSEEL 2019). The MSEEL is a
multi-institutional, long-term collaborative field site where integrated geoscience,
engineering, and environmental research have been conducted to assess
environmental impacts and develop new technology to improve recovery efficiency as
well as reduce environmental footprint of shale gas operations (MSEEL 2019).

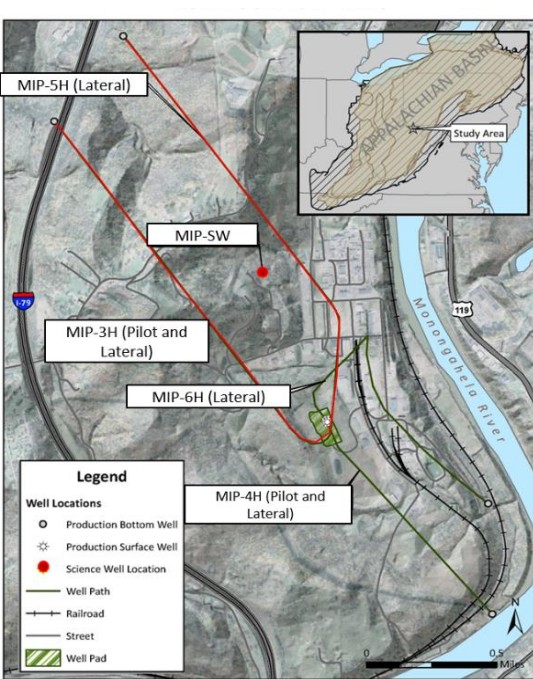

Figure 1. Location of the Marcellus Shale Energy and Environment
Laboratory and the four production wells.





The MSEEL is the site of two horizontal production wells completed in 2011 (wells
4H and 6H, Figure 1) and two horizontal production wells completed in 2015 (wells
3H and 5H, Figure 1).  Production from the newer horizontal wells began in
December 2015.  Dates and duration for phases of operation are shown in Figure S2.
The vertical drilling was conducted using three diesel Caterpillar 3512 engines with
1365 kW generators.  Horizontal drilling made use of two dual fuel (40% diesel and
60% natural gas) engines.  All activities at the well pad followed industry's best
management practices (MSEEL 2019).
**Air Quality and Meteorological Data Collection**
An ambient air monitoring laboratory (18' trailer with ambient air sampled from inlets
on the trailer roof) was situated at the northeastern corner of the MSEEL well pad
(Figure 1).  With wind direction at this location most frequently from the southwest
(Figure 2), this position optimized the occurrences of the laboratory being downwind
of the well pad.  Instrumentation in the laboratory and measured constituents are listed
in Table 1.  All instruments were maintained and calibrated according to
manufacturer's recommended protocols.  Details of the laboratory assembly and
operation have been previously described (Pekney et al. 2014).
Data collected at the air monitoring site is classified by activity at the well pad.
Horizontal drilling occurred September 8 – October 5, 2015, first at well 5H then at
well 3H.  Hydraulic fracturing occurred October 10 – November 16. Cleanout
activities followed on November 20-26, which involved using a diesel-powered coil
tubing rig to drill out plugs and flush out residue left in the wells.






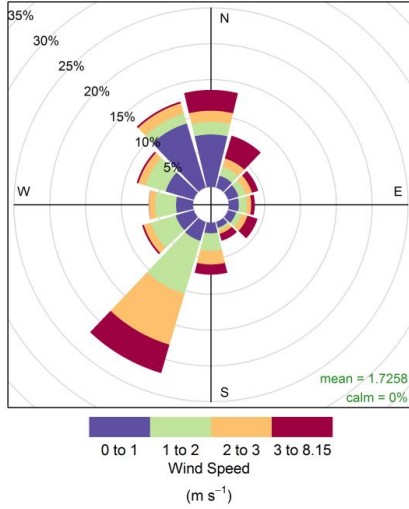

Figure 2. Wind speed and direction during ambient air monitoring campaign at MSEEL (September 2015-February 2016).

Flowback, the flowing of gas, formation fluid, and frac fluid up the wells to the
surface, took place over December 10-14, after which both wells were in production.
A reduced emission completion (REC) was performed; gas produced during this time
was captured using portable equipment brought on site that separates the gas from the
liquids so that the gas can be retained as a product.
Air monitoring began September 18, 2015 and ended February 1, 2016. No data were
collected for the vertical drilling phase.  Data collection was continuous except for
calibration and instrument downtime.  The laboratory's meteorological station
measured relative humidity, temperature, rainfall, solar radiation, wind direction,
wind speed, and barometric pressure at an elevation of 10m.






Table 1. Constituents measured by the MSEEL mobile air monitoring laboratory
(Pekney et al. 2018).

| Measurement | Unit | Resolution | Sampling Rate | Instrument | Measurement technique |
|---|---|---|---|---|---|
| VOCs (52 compounds, see Table S1 for full list) | ppb | 0.4 ppb | 1 hour | Perkin Elmer Ozone Precursor Analyzer (Waltham, Massachusetts) | Gas Chromatograph with Flame Ionization Detection (GC—FID) with thermal desorption |
| Ozone, $NO_x$ | ppb | 0.4 ppb Ozone, 50 ppb $NO_x$ | 1 minute | Teledyne-API Gas Analyzers T400 and T200U (San Diego, California) | UV absorption, Chemiluminescence |
| Methane, carbon dioxide | ppm | <5 ppb Methane, 1 ppm $CO_2$ | 1 second | Picarro G2201-i (Santa Clara, California) | Cavity Ring-Down Spectrometry |
| Meteorological Parameters: wind speed and direction, temperature, relative humidity, barometric pressure, rainfall, and solar intensity | various | Various; 1 degree for wind direction/ 0.45 m/s for wind speed for Vantage Pro2 Plus; 0.1 degree for wind direction/ 0.01 m/s wind speed for R.M. Young 81000 | 1 minute | Davis Instruments Vantage Pro2 Plus (Oakland, California) and R.M. Young 81000 ultrasonic anemometer (Traverse City, Michigan) | Various |

**Source-Receptor Modeling**
Positive Matrix Factorization (PMF), a factor analysis method (Figure S3), was
applied to hourly averaged ambient concentrations of measured species to identify
possible sources and patterns for the stages of development.  PMF decomposes the
sample data into two matrices: factor profiles (representative of *sources*) and factor
contributions (Brown et al. 2015; Norris et al. 2014).  The fundamental objective of
PMF is to solve the chemical mass balance (Equation 1) between measured species
concentrations and source profiles while optimizing goodness of fit (Equation 2):





Mass balance (Evans and Jeong 2007):
$$x_{i,j} = \sum_{k=1}^{p} g_{ik} f_{kj} + e_{ij}$$

[1]

where $x_{i,j}$ is the data matrix with dimensions of $i$ (observations) by $j$ (chemical
species), $p$ is the optimum number of factors, $g_{ik}$ is the factor contribution to the
observation, $f_{kj}$ is the species profile of the factor, $k$ is the factor, and $e_{i,j}$ is the residual
concentration for each observation.
Goodness of fit:
$$Q = \sum_{i=1}^{n} \sum_{j=1}^{m} \left( \frac{x_{ij} - \sum_{k=1}^{p} g_{ik} f_{ij}}{s_{ij}} \right)^2$$

[2]

where $Q$ is the goodness of fit, $n$ is the total number of observations, $m$ is the total
number of chemical species, and $s_{ij}$ is the uncertainty for each observation. Summary
of methods for uncertainty calculations are provided in Supplemental Information.
Missing values within the data set are replaced with the median value of that species;
also, uncertainty for missing values is set at four times the species-specific median by
the program. Multiple runs with different numbers of factors are executed for each
data set. The output of the PMF analysis needs to be interpreted by the user to identify
the number of factors that may be contributing to the samples and the possible sources
they represent.  One of the main strengths of PMF     analysis is that each sample is
weighted individually, which allows the user to adjust the influence of each sample
based on the measurement confidence.
Signal-to-noise ratio (S/N), an indicator of the accuracy of the variability in the
measurements, can be used to identify a species as "Strong", "Weak", or "Bad".





Generally, if this ratio is greater than 0.5 but less than 1 that species has a "Weak"
signal. "Strong" is the default value for all species with an assumption of S/N greater
than 1. "Bad" category excludes the species from the rest of the analysis. We
considered the number of samples that are missing or below the detection limit when
choosing the category for each species. (Norris et al. 2014). The expected goodness
of fit ($Q_{expected}$) is calculated for each scenario (Norris *et al.*, 2014):
Expected goodness of fit:
$$Q_{expected} = (i \times j) - \{(p \times i) + (p \times j)\}$$

[3]

where (i x j) is the number of non-weak data values in Xij and (p x i) and (p x j) are
the number of elements in G and F, respectively. $Q_{robust}$ is the calculated goodness-of-
fit parameter that excludes points that are not fit by the model. The lowest
$Q_{robust}/Q_{expected}$ is calculated to compare different factor scenarios; when changes in Q
become small with increasing factors, it can indicate that there may be too many
factors in the solution (Brown et al. 2015).
In addition to these calculated parameters, factor profiles and error estimation
diagnostics are used to compare the output of different simulations. Marker species
(chemical species that are unique to a particular source) and temporal or seasonal
variations can be used to aid in identifying the possible emission sources (Figure 3).
Associations between factors can also provide useful information for profile
characterization. Moreover, meteorological data can provide useful information about
the geographic location of the sources.
In order to perform the PMF analysis, we utilized a user-friendly graphical user
interface (GUI) developed by the U.S. Environmental Protection Agency (EPA), EPA
PMF 5.0 (Norris et al., 2014). Hourly average data was used for each pollutant to





unify the measurement intervals. All pollutants included in the matrix were identified
as "strong" (signal to noise: S/N > 2). Fifty base runs were performed, and the run
with the minimum Q value was selected as the base run solution. In each case, the
model was run in the robust mode with a number of repeat runs to ensure the model
least-squares solution represents a global rather than a local minimum. First, the
rotational (linear transformation) Fpeak variable was held at the default value of 0.0.
However, there can be almost infinite possibilities of F and G matrices that produces
the same minimum Q value, but the goal is producing a unique solution. As a result,
rotational freedom is one of the main sources of uncertainty in PMF solutions
(Paatero et al. 2014). Therefore, Fpeak values were adjusted (-1.0, -0.5, 0.5, and 1.0)
to explore how much rotational ambiguity exists in PMF solutions. In other words, the
model adds and/or subtracts rows and columns of F and G matrices based on the
Fpeak value, which is typically between -5 and +5 (Norris et al. 2014). Positive Fpeak
values cause a sharpened F-matrix and smeared G-matrix; negative Fpeak values
result in subtractions in the G-matrix. The factor contributions were analyzed to find
the optimum Fpeak value.
The PMF analysis was completed with error estimation. We used three methods of
error estimation: Bootstrap (BS), Displacement (DISP), and BS-DISP, which guide
understanding the stability of the PMF solution (Norris et al. 2014). BS analysis is
used to determine whether a set of observations affect the solution disproportionately.
The main idea of BS analysis is resampling different versions of the original data set
and perform PMF analysis. Random errors and rotational ambiguity affect BS error
intervals. The main reason of rotational ambiguity is the existence of infinite solutions
similar to the solution generated by PMF solution. DISP analysis helps to analyze the
PMF solution in detail. Only rotational ambiguity affects DISP error intervals.
BS-DISP is a hybrid method that gives more robust results than DISP results.



**Results and Discussion**

**Overview of Results for Measured Compounds**

Figure 3 shows a box-and-whisker graph of the measured $NO_x$, NO, $NO_2$, Ozone, and

ethane during the whole monitoring campaign at the study site. Similarly, Figure 4

shows a statistical summary of methane and carbon dioxide. The y-axis represents

concentrations and the x-axis represents the phases of the well development. The

black line on each of the boxes represents the median for that particular data set. The

small circles represent outliers. The blue circles represent the mean. Since most of the

VOCs concentrations measured were consistently below 10 ppb, only ethane is

included. There was an increase for $NO_x$ ($25^{th}$ percentile (q1)=12.5 ppb) and NO (q

1= 2.7 ppb) during the *fracturing* phase compared to other phases. The whiskers show

the high variability for this phase, which can be a result of small sample size for the

*fracturing* phase. $NO/NO_2$ ratio for $25^{th}$ and $75^{th}$ percentiles was 1.2, indicating

fresher, less oxidized emissions. The skewness of the data for this phase indicates that

the data may not be normally distributed. $NO_2$ graph shows a similar trend for the

*fracturing* phase. We did not observe significant differences for different development

phases for ozone, which is not surprising as it is a secondary pollutant and it can be

related to winter season of the data collection period. (Peter M. Edwards et al. 2014).

There was a dramatic increase for the flowback phase for ethane concentration. This

$25^{th}$ percentile was 24 ppb, while this concentration ranged between 0 and 11 ppb for

other phases. The $75^{th}$ percentile was 89 ppb, which is a significantly higher value

compared to other phases. We observed a similar trend for methane concentration.

The $25^{th}$ percentile (2.5 ppm) and the $75^{th}$ percentile (4.3 ppm) were significantly

higher than other phases. Differences for development phases for $CO_2$ were not

statistically significantly different.  $CO_2$ has many emissions sources and variable



background concentrations so distinguishing emissions from the well pad activities is
difficult.

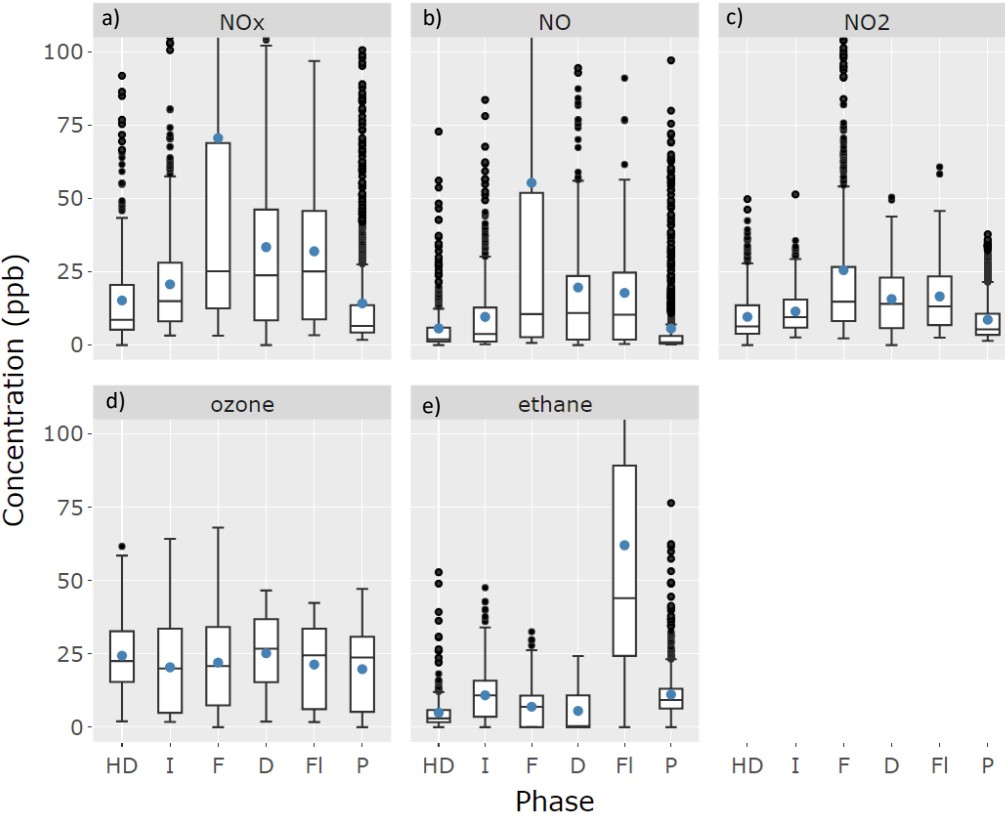

Figure 3. Summary statistics of input parameters for (a) NOₓ, (b) NO, (c) NO₂, (d) Ozone, € ethane
(HD: Horizontal Drilling, I: Idle, F: Fracturing, D: Drillout, Fl: Flowback, P: Production. The idle
phase consists of gaps of time between other operational phases, when there was little to no
emissions-generating activity on the well pad.

The average concentrations of methane and ethane for the entire monitoring campaign
are shown in Figure S4. The highest ethane concentrations occurred during the
*flowblack* stage (565.7 ppb).  A mean that is significantly higher than the median
comes from a distribution that is skewed due to peak events (mean$_{ethane}$= 11.4 ppb,
median$_{ethane}$= 8.5 ppb). Propane and isobutane had the second and third highest
average concentrations, respectively, for each phase of development. Similarly, the



hourly concentration graphs of $NO_x$, $O_3$, and $CH_4$, and $CO_2$ were used to analyze the
factor solutions (Figure S5).
**Factor Profiles**
The three-factor model was chosen for the PMF analysis based on the interpretation
of the factor profiles, $Q_{robust}/Q_{expected}$ ratios, factor contributions, error estimation
results, and hourly peak concentrations of pollutants (Figure S6). The three-factor
solution was resolved to the following factors: *natural gas* for the natural gas-related
emissions sources; *regional transport/photochemistry* for the atmospheric regional
molecular transport and oxidized background air; and *engine emissions* for emissions
from vehicles, drill rigs, generators, and pumps used at the site (Figure 5). The
summary of PMF models with various Fpeak values for well development activities
are shown in Table S4. The DISP, BS, and BS-DISP results for 2, 3, and 4 factor PMF
solutions are summarized in Table S2. For the 3-factor analysis, the DISP results
indicate that there are no swaps and the PMF solution is stable, which means there are
no exchange factor identities and it is a well-defined solution for the case. According
to BS results, there is a small uncertainty; this can be an impact of high variability in
concentration. BS-DISP captures both random errors and rotational ambiguity; these
results also indicate that the solution is reliable because there are no swaps between
factors for the PMF model. Error estimation summary plots (Figure S6) show range of
concentration by species in each factor: Base Value, BS 5th, BS Median, BS 95th,
BS-DISP 5th, BS-DISP Average, BS-DISP 95th, DISP Min, DISP Average, and DISP
Max.


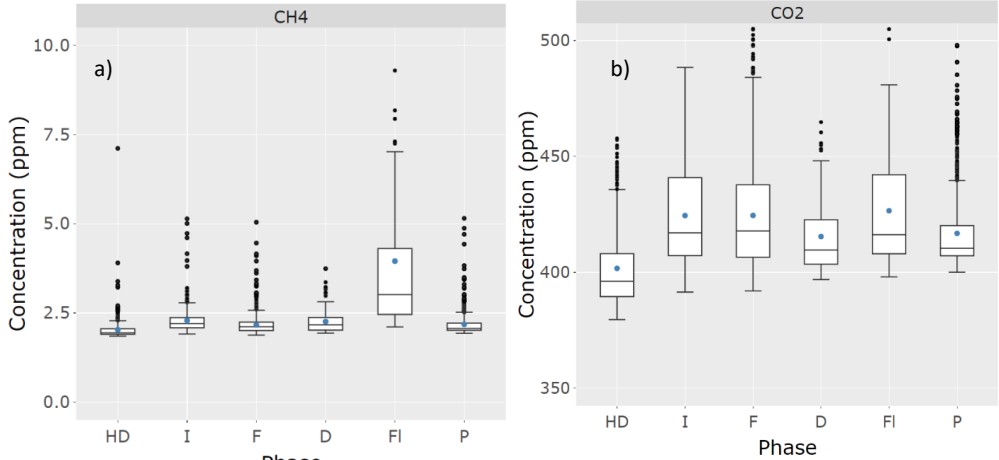

Figure 4. Summary statistics of input parameters for methane (a) and carbon dioxide
for (b) (HD: Horizontal Drilling, I: Idle, F: Fracturing, D: Drillout, Fl: Flowback, P:
Production.
**Source Profiles**
**The natural gas factor** was named as such due to its composition of species that are
present in natural gas:  1% methane, 3% ethane, 1.5% propane, 0.5% isobutane, 1% n-
butane, 0.1% pentane, and 0.2% isopentane.  Ethane is a particularly good marker for
natural gas emissions sources due because its atmospheric sources are almost
exclusively from natural gas extraction, production, processing and use (Liao et al.
2017). Ninety-two percent of ethane mass is explained by the natural gas factor.  The
highest contribution for this factor occurred during the flowback phase.
**The regional transport/photochemistry factor** was characterized by high
contributions from ozone (12%), $CH_4$ (1%), and $CO_2$ (86%). Ninety-nine percent of
the ozone mass was explained by this factor.  Ozone is a product of photochemistry
and not directly emitted by any of the sources on the well pad.  Although $CH_4$ and
$CO_2$ would be emitted by well pad sources, they are also present in background
ambient air and could be transported to the monitoring location from other sources in



the region.  Contributions of this factor were relatively steady for all phases of
operation during the entire monitoring campaign.
**The engine emissions factor** was composed of 39% NOx, 33% NO, and 11% $NO_2$ as
well as 0.02% toluene and 0.04% benzene.  The portions of the mass of these species
explained by this factor are 74%, 87%, 60%, 20%, and 54%, respectively.  Toluene is
released mainly from motor vehicle emissions and chemical spills (Gierczak et al.
2017). Contribution of this factor was significantly highest during hydraulic
fracturing, when there were emissions from many diesel engines operating
continuously on the well pad.  Contribution during flowback was also elevated.
Several peaks of contribution were observed during production, which could be due to
maintenance vehicles and other short-lived vehicle-based activities on the well pad.

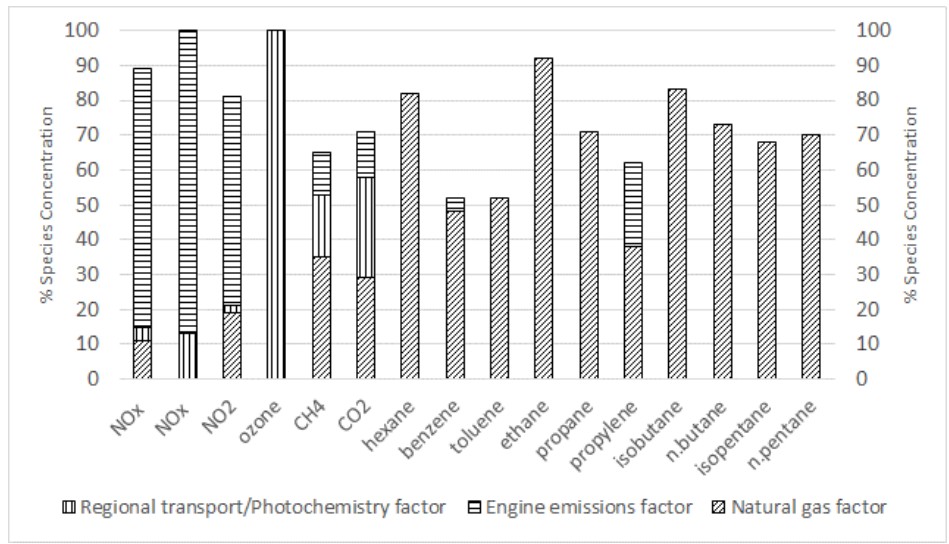

Figure 5. The three-factor solution fingerprints for Drilling through Production
Monitoring Period, $F_{peak}$=1.
The main limitation of the study is having uneven number of data points for each
operational phase. This limitation affects the analyses; however, we do not have
control of the durations of the operational phases. As a future work, integrating more
data from different fields can decrease the inherent uncertainty.



**Conclusion**

We investigated the effect of unconventional natural gas development activities on

local air quality by using ambient air monitoring laboratory near Marcellus Shale well

pad in Morgantown, Western Virginia. The results of PMF solutions for well pad

development phases show that there were three potential factor profiles as outlined in

Figure 5: *natural gas*, *regional transport/photochemistry*, and *engine emissions*.

Horizontal drilling stage had an important contribution to the *natural gas* factor. In

addition, there was a significant concentration contribution at the end of the horizontal

drilling phase. An increasing contribution to *engine emission* factor was observed

over different well pad drilling through production phases. The peak concentration

was observed during the drillout stage. Even though it is difficult to compare the

*regional transport/photochemistry* contributions due to high variability, highest

contributions occurred during horizontal drilling and drillout.

As determined by the PMF analysis, a measurable increase in natural gas-related

pollutant concentrations and the associated natural gas factor contribution from

different stages of active phase was not observed. At the downwind distance of 600m

from the well pad center to the air monitoring laboratory, the emissions from the well

pad were not easily distinguishable from typical variations in ambient background

concentrations.  West Virginia has many natural gas wells that contribute to the

ambient background, as evidenced by ethane concentrations that are higher than

typical global background (Rinsland et al. 1987; Rudolph et al. 1996).  Short-lived

peak events that were observed when the wind direction was coming from the well

pad show that emissions can be dispersed downwind and detected at this distance, but

when concentrations are averaged and analyzed with a PMF analysis the peak events

were not significant enough to result in a measurable impact of the well pad emissions

at the receptor location. Understanding the air quality impacts of operational phases is





important since it has potential to help inform future decision-making and constrain
cumulative impact assessments.

**Conflicts of interest**
There are no conflicts to declare.
**Acknowledgements**
Disclaimer: This report was prepared as an account of work sponsored by an agency
of the United States Government. Neither the United States Government nor any
agency thereof, nor any of their employees, makes any warranty, express or implied,
or assumes any legal liability or responsibility for the accuracy, completeness, or
usefulness of any information, apparatus, product, or process disclosed, or represents
that its use would not infringe privately owned rights. Reference therein to any
specific commercial product, process, or service by trade name, trademark,
manufacturer, or otherwise does not necessarily constitute or imply its endorsement,
recommendation, or favoring by the United States Government or any agency thereof.
The views and opinions of authors expressed therein do not necessarily state or reflect
those of the United States Government or any agency thereof.
This technical effort was performed in support of the National Energy Technology
Laboratory's ongoing research under the Natural Gas Infrastructure Field Work
Proposal DOE 1022424. This research was supported in part by appointments to the
National Energy Technology Laboratory Research Participation Program, sponsored
by the U.S. Department of Energy and administered by the Oak Ridge Institute for
Science and Education. Authors would also like to thank James I. Sams III, and
Richard W. Hammack.
**Author Contribution**



**Nur H Orak:** Conceptualization, Methodology, Software. Visualization, Writing
**Natalie J. Pekney:** Supervision, Methodology, Writing. **Matthew Reeder:**
Methodology, Validation.
**Code/Data availability**
Model simulations presented in this paper are available upon request.

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
