# Peer review of "Identifying and Quantifying Source Contributions of Air Quality Contaminants"

_Atmospheric Chemistry and Physics, 2020_

## Author Comment (AC1) · 20 Oct 2020

Q: The authors should be made aware of this report that is of similar scope and should be a useful comparison.

https://www.colorado.gov/airquality/tech_doc_repository.aspx?action=open&file=CSU_NFR_Report_Final_20160908.pdf
A: We would like to thank the reviewer for the suggestion, we will compare the results.
Q: "The objective of this study is to investigate the effect of unconventional natural gas development activities on local air quality as observed at an active Marcellus Shale well pad" it would seem the objective is better stated as investigating the "emissions at the well pad", not the effect on local air quality, which is still interesting,

but different. "Moreover, model results suggest that the major contributions to the regional transport/photochemistry factor occurred during horizontal drilling and drillout stages." This is just a relative shift where there are less emissions at the site so the background factor appears more pronounced. As it reads, one could think it was contributions to photochemistry or nearby photochemistry–worth clarifying. A: We regret for the confusion, we will rewrite the sentences. Q: Fig 3 caption seems to end abruptly. Also, adding dates in the caption for each phase would be extremely helpful. A: We regret for the mistake. We corrected the format of the Figure 3 caption. There is not enough space for the dates. If we add the dates, it would be difficult to read the figure. Instead, we shared the unconventional natural gas production process activity diagram with dates in the SI.

Q: The SI states there is a lot more VOC data, anything interesting in there? Are there indeed significant aromatics in the Marcellus natural gas emissions? A: We have analyzed all VOCs before deciding the parameters for the PMF model. The most significant compounds are included in the study. The rest of the VOCs are listed in the SI but excluded in the model.

---

## Editor Comment (EC1) · Thomas Karl (Editor) · 17 Nov 2020

Dear authors, thank you for posting your response to the quick reviews of this manuscript. Please be aware that the actual reviews are pending and will be posted soon. Thomas Karl

---

## Referee Comment (RC1) · Anonymous Referee #2 · 23 Nov 2020

The manuscript examines concentrations of a range or pollutants (e.g. CH4, CO2, O3, VOCs, NO, NO2) using a monitoring laboratory near a well-pad to examine emissions pathways using PMF during horizontal drilling through production phases. The study reports 2 factors related to the on-site operations: a "natural gas" factor and an "engine emission" factor. It is an interesting study, but it would benefit greatly from additional discussion of the results and what they mean for the relative importance of the emission pathways for the various pollutants observed. Given the importance of methane, this would be an especially important place to dive in further. For example, over 3 pages are spent describing well-established details of the PMF method and application, but only 1.5 pages are spent on the source profile results themselves (including 1 figure).

This is not to say that the length of a discussion is indicative of quality, but it is provided here as an example of where the interpretation of results are relatively thin compared with the discussion of methods and the reporting of data. As another example, the manuscript closes with "Understanding the air quality impacts of operational phases is important since it has potential to help inform future decision-making and constrain cumulative impact assessments" and it would seem like the authors have data here to say something more quantitative about the different operational phases and their relative contributions, but the discussion and analysis does not sufficiently touch on this (and would need to account the wind issues identified below).

Overall, the paper has the potential to make useful contributions to the field, but should be carefully reviewed by the authors to expand on the discussion of their results where possible. A range of other comments are provided below.

Abstract:

- "The objective of this study is to investigate the effect of unconventional natural gas development activities on local air quality as observed at an active Marcellus Shale well pad" it would seem the objective is better stated as investigating the "emissions at the well pad", not the effect on local air quality, which is still interesting, but different. Please consider revising.

- "Moreover, model results suggest that the major contributions to the regional trans-port/photochemistry factor occurred during horizontal drilling and drillout stages." This is just a relative shift where there are less emissions at the site so the background factor appears more pronounced. As it reads, one could think it was contributions to photochemistry or nearby photochemistry–worth clarifying.

Fig 1: where is the monitoring lab on this map?

Lines 259-262 discuss a lot of basic details about the figure that would be better in the caption, with the results and discussion section focused on substantive observations.

Fig 3 caption seems to end abruptly. Also, adding dates in the caption for each phase would be extremely helpful.

An opportunity is missed to comment on the production volumes of the well and how that relates to the observed "natural gas" factor emissions. There could also be more discussion of this factor. Efforts like this to quantify emissions would be useful, or at least to discuss quantitatively the role of each factor in the observed emissions (2 of the factors)–this only exists in fig 5.

Fig 5 could be much more clear. I suggest incorporating color.

It is unclear how the wind direction is driving the PMF results and the observed average concentrations of pollutants at the site (Figs 3-4). The authors state that "With wind direction at this location most frequently from the southwest (Figure 2), this position optimized the occurrences of the laboratory being downwind of the well pad." But the wind is still only coming from that direction ~23% of the time and there is no clear understanding of how that changes over the course of the study. Some effort needs to be taken to demonstrate that this is not biasing the concentrations or PMF results, perhaps by conducting the PMF analysis with the wind isolated to that direction, or quadrant. Otherwise the wind direction could be a driver of relative differences in the PMF factors.

- For example, it could play a role in observations like this "The skewness of the data for this phase indicates that the data may not be normally distributed." (line 268)

The SI states there is a lot more VOC data, anything interesting in there? Are there indeed significant aromatics in the Marcellus natural gas emissions as the data would suggest or is this just from the engine exhaust factor? Fig 5 shows it all on the natural gas factor, but the text talks about toluene with the engine factor. This is unclear and needs to be cleared up.

- Can the VOC data be used to further substantiate the observed PMF factors (beyond

what is done so far, which is useful)? This is a hint of this that appears at line 317. This seems like a big data set, but most of it is just left to the list in the SI without any data.

- "Propane and isobutane had the second and third highest average concentrations, respectively, for each phase of development." (line 286), might be useful to clarify and show a figure in the SI for this since the sentence is not fully clear.

Supplement:

Generally, this needs to be cleaned up with more. Some of the figures are hard to read and the final section on uncertainty estimation is very challenging to follow given its structure.

Figure S4-5 could be quite useful, even in the main text if the format showed the periods of the different activities with sufficient image clarity.

Figure S6 needs a better caption to explain what is being shown.
* * *

---

## Referee Comment (RC2) · Anonymous Referee #1 · 26 Nov 2020

The authors present several months of near-continuous monitoring of a suite of air quality metrics including NOx, CO, CO2, CH4, and VOCs at site located near a natural gas well in the Marcellus Shale region of West Virginia. The measurements were conducted throughout the key phases of the drilling process, including hydraulic fracturing and flowback of a horizontally-drilled well. The authors present the observed mixing ratios for select species during different well-activity periods and use positive matrix factorization (PMF) to identify "three factors impacting air quality at the site" which basically boil down to the background air, flowback/natural gas fugitive emissions, and engine exhaust.

The measurements are valuable but the analysis could use a more thoughtful/through approach and attention paid to similar studies that were not cited. I recommend publications after major revisions/additions.

General comments:

Measurement details – there isn't much information included on the distance/direction between the measurement site and the drilling activities. Other than a wind rose (Figure 2), how often was the measurement site directly downwind? Do any of the PMF factors or other chemical parameters relate to the windspeed and direction?

The authors left a lot of useful analysis out regarding the VOC measurements, which are a key component of air quality measurements and are relatively scarce in the literature making these all the more important to expand upon. The authors should report the ethane/methane ratio for comparison (see Yacovitch et al. 2014) as it is a key metric to defining natural gas emissions from other methane sources and is important for emissions modeling purposes. Also, the iso- to n-pentane ratio that is used to separate gasoline related sources from raw oil and natural gas (see Gilman et al. 2013). Gasoline/traffic is an important source that I'm surprised the PMF didn't pick out as the measurement site looks to be in close proximity to an interstate highway. The basic statistics of all VOCs should be reported for the flowback portion at a minimum.

Lastly, this isn't the first study of its kind (L117). Please refer to and include discussion of Hecobian et al. (2019). Whenever possible, please use the correct engineering terms such as "hydraulic fracturing" instead of "frac" or "fracking" to be more precise.

Other comments in order of appearance:

Line 72: Simulated what? Drilling, the economy, …?

Line 73: By the public of the potential public health impacts…

Line 74: hydraulic fracturing, referred to as "fracking"

Line 82: Add Hecobian et al. 2019

Line 97:  Add reference Gilman et al. 2015

Line 117:  Hecobian et al. 2019.  Not the first/only but likely so if limiting to Marcellus.

Line 157: hydraulic fracturing fluid

Line 270:  Did you see any instances of NOx titration?  Often, with sharp NO spikes in concentration from local sources, you will see an equally sharp decrease in ozone.

Line 272:  The reference to Edwards et al. takes on several different forms throughout the manuscript – be consistent.

Line 317:  How is natural gas only 1% methane???  This doesn't make any sense.  Also, you expect the n-alkane isomers to be more prevalent in natural gas than the branched isomers; however, you are reporting iso-pentane > n-pentane which sounds more like a mobile source emission.  These percentages aren't consistent with other oil and natural gas studies.  How does this composition compare to the Swarthout paper or any other source in the Marcellus?

Line 323:  This sounds more like a regional background.  How do you know it's transport or active chemistry?

Line 334:  Toluene is also a known component of oil and gas extraction and is often in hydraulic fracturing fluid.

References:

**Demonstration of an Ethane Spectrometer for Methane Source Identification**.  T. I. Yacovitch, S. C. Herndon, J. R. Roscioli, C. Floerchinger, R. M. McGovern, M. Agnese, G. Petron, J. Kofler, C. Sweeney, A. Karion, S. A. Conley, E. A. Kort, L. Naehle, M. Fischer, L. Hildebrandt, J. Koeth, J. B. McManus, D. D. Nelson, M. S. Zahniser and C. E. Kolb. *Environmental Science & Technology*, 48(14), 8028-8034, doi:10.1021/es501475q, **2014**

**Source Signature of Volatile Organic Compounds from Oil and Natural Gas Operations in Northeastern Colorado**.  J. B. Gilman, B. M. Lerner, W. C. Kuster and J. A. de Gouw.  *Environmental Science & Technology*, 47(3), 1297-1305, doi:10.1021/es304119a, **2013**

**Air Toxics and Other Volatile Organic Compound Emissions from Unconventional Oil and Gas Development**.  A. Hecobian, A. L. Clements, K. B. Shonkwiler, Y. Zhou, L. P. MacDonald, N. Hilliard, B. L. Wells, B. Bibeau, J. M. Ham, J. R. Pierce and J. L. Collett.  *Environmental Science & Technology Letters*, 6(12), 720-726, doi:10.1021/acs.estlett.9b00591, **2019**

---

## Author Response (AR1)

Dear Prof. Karl,

We are grateful to the editor and reviewers for their time and constructive comments on our manuscript. We have implemented their comments and suggestions and wish to submit a revised version of the manuscript for further consideration in the journal. The objective of the Marcellus Shale Energy and Environment Laboratory (MSEEL) is to provide a long-term field site to develop and validate new knowledge and technology to improve recovery efficiency and minimize environmental implications of unconventional resource development. Therefore, there are several publications from the MSEEL data and some of them are in progress, which will answer some of the questions and suggestions raised by reviewers. The aim of this manuscript is not analyzing the trends of each air pollutant but exploring the source of each emission during different phases of unconventional natural gas well pad development by Positive Matrix Factorization method. We have implemented major revisions to be able to answer all questions by the reviewers. Changes in the initial version of the manuscript are highlighted (with 'tracked changes') for added sentences or strikethrough for deleted sentences in the revised version. Below, we also provide a point-by-point response explaining how we have addressed each of the reviewers' comments. We look forward to the outcome of your assessment.

Yours sincerely,

On behalf of the co-authors

Nur H. Orak, Ph.D.

Assistant Professor
Department of Environmental Engineering
Marmara University
Istanbul, Turkey

**"Identifying and Quantifying Source Contributions of Air Quality Contaminants during Unconventional Shale Gas Extraction" by Nur H. Orak et al.**

*\* Comments from Referees are in black, authors' responses are in green, changes are in red color.*

**Comments from Reviewer #1**

The authors present several months of near-continuous monitoring of a suite of air quality metrics including NOx, CO, CO2, CH4, and VOCs at site located near a natural gas well in the Marcellus Shale region of West Virginia. The measurements were conducted throughout the key phases of the drilling process, including hydraulic fracturing and flowback of a horizontally-drilled well. The authors present the observed mixing ratios for select species during different well-activity periods and use positive matrix factorization (PMF) to identify "three factors impacting air quality at the site" which basically boil down to the background air, flowback/natural gas fugitive emissions, and engine exhaust.

The measurements are valuable but the analysis could use a more thoughtful/through approach and attention paid to similar studies that were not cited. I recommend publications after major revisions/additions.

A: Authors would like to thank reviewer #1 for the detailed review, critiques, and suggestions. Authors' responses are under each comment and a revised manuscript is provided with marked-up changes.

General comments:

**Q1.** Measurement details – there isn't much information included on the distance/direction between the measurement site and the drilling activities. Other than a wind rose (Figure 2), how often was the measurement site directly downwind? Do any of the PMF factors or other chemical parameters relate to the windspeed and direction?

A:

We have revised Figure 1 and merged with a second figure that shows the location of the trailer with respect to the location of the wells and the boundaries of the well pad. We have added the following explanation: "Figure 1 shows the location of the trailer with respect to the location of the wells and the boundaries of the well pad. The distance between the wells and the trailer is 90 m."

[Figure]

Figure 1. Location of the Marcellus Shale Energy and Environment Laboratory and the four production wells.

Wind rose data is specific to each activity but we did not change the location of the mobile laboratory during data collection. First, we developed several PMF models and analyzed the results before deciding the final PMF model that is represented in the manuscript. We did not explain every single PMF run on this manuscript, which would result more than 50 pages. We presented the final PMF model. Wind direction does not serve as an input to the PMF analysis. We can only add a variable that we are analyzing the source of emission. We have tried to use TEOM data for the preliminary models but these parameters are excluded for the final model due to limited contribution to the PMF. Most of the sources on the site released emissions near ground level. With the monitoring laboratory located on the pad but on the downwind edge, it is assumed that sufficient dispersion occurred to detect emission events.

**Q2.** The authors left a lot of useful analysis out regarding the VOC measurements, which are a key component of air quality measurements and are relatively scarce in the literature making these all the more important to expand upon. The authors should report the ethane/methane ratio for comparison (see Yacovitch et al. 2014) as it is a key metric to defining natural gas emissions from other methane sources and is important for emissions modeling purposes. Also, the iso-to n-pentane ratio that is used to separate gasoline related sources from raw oil and natural gas (see Gilman et al. 2013).

A: The Marcellus Shale Energy and Environment Laboratory (MSEEL) is a long-term project that provides and validates new knowledge to minimize environmental impacts of unconventional resource development. Therefore, there are several studies that are published based on the MSEEL database, you can see the list of publications on http://mseel.org/research/publications.html. In this study, our aim is not providing a detailed

analysis of the time series of concentrations of VOCs or other parameters. This will be provided in another publication currently in progress. In this study, we aim to investigate the emissions at different well pad activities by PMF as an alternative method. However, we have conducted a detailed analysis of VOCs before selecting the most significant compounds for the PMF model. We have run several PMF analysis to compare different subset of data, but results did not give any meaningful results to report. To be able to answer your questions, we have added a new table in SI to show the average concentrations for VOCs; Table S2. Average concentrations (ppb) of the most significant volatile organic compounds in different operational phases. Since most of the VOCs concentrations measured were consistently below 10 ppb, only ethane is included.

**Q3.** Gasoline/traffic is an important source that I'm surprised the PMF didn't pick out as the measurement site looks to be in close proximity to an interstate highway. The basic statistics of all VOCs should be reported for the flowback portion at a minimum.

A: Engine emissions was one of the factors identified by PMF. Because of the proximity to the highway and other roads (this is an urban location) the model was unable to distinguish between engine emissions from the highway or other nearby road from on-well pad engine emissions. We have added the average concentrations of top ten VOCs in different operational phases in Table S2. In addition, we have added a new figure in the main manuscript to show the ratio of ethane to methane during different operational phases (Figure 4).

**Q4.** Lastly, this isn't the first study of its kind (L117). Please refer to and include discussion of Hecobian et al. (2019). Whenever possible, please use the correct engineering terms such as "hydraulic fracturing" instead of "frac" or "fracking" to be more precise.

A: Thank you for the suggestion. We have referred Hecobian et al. 2019 on Line 81.

In the original abstract, we have mentioned the short form of hydraulic fracturing as "fracking" as follow: "…to the technological development of hydraulic fracturing ("fracking")." and we have used the term throughout the manuscript. However, in the revised manuscript, we have revised all "fracking" terms as "hydraulic fracturing" whenever possible.

**Q5.** Other comments in order of appearance:

Line 72: Simulated what? Drilling, the economy, …?

A: We have revised the sentence as follow: "…has stimulated drilling activities in other countries."

Line 73: By the public of the potential public health impacts…

A: We regret for the mistake and revise the sentence as follow: "…by the public for the…"

Line 74: hydraulic fracturing, referred to as "fracking"

A: Thank you for the suggestion, we have revised the term here and also throughout the manuscript.

Line 82: Add Hecobian et al. 2019

A: Thank you for the suggestion, we have added the reference.

Line 97: Add reference Gilman et al. 2015

A: Thank you for the suggestion, we have added the reference.

Line 117: Hecobian et al. 2019. Not the first/only but likely so if limiting to Marcellus.

A: We regret for the mistake, we have revised the sentence as follow: "This is the first study, to our knowledge, to collect high time resolution ambient concentrations of compounds emitted from well pad activity on Marcellus Shale during various phases of operation such that the relative air quality effect of each phase of development can be investigated."

Line 157: hydraulic fracturing fluid

A: Thank you for the suggestion, we have revised the sentence.

Line 270: Did you see any instances of NOx titration? Often, with sharp NO spikes in concentration from local sources, you will see an equally sharp decrease in ozone.

A: We have examined the peaks of NOx data but we have not seen any significant changes in ozone concentrations.

Line 272: The reference to Edwards et al. takes on several different forms throughout the manuscript – be consistent.

A: We regret for the mistake and corrected the format.

Line 317: How is natural gas only 1% methane??? This doesn't make any sense. Also, you expect the nalkane isomers to be more prevalent in natural gas than the branched isomers; however, you are reporting iso-pentane > n-pentane which sounds more like a mobile source emission. These percentages aren't consistent with other oil and natural gas studies. How does this composition compare to the Swarthout paper or any other source in the Marcellus?

A: This is not the concentration of composition species in methane, this statement addresses the data percentage of species that feed natural gas factor. The PMF model provides three factors with different compositions, we analyze the composition to find the main source of the factor. For example, here it is natural gas due to high ethane contribution to the factor.

For clarification, we have added a new figure for factor profiles in SI as you can see below.

[Figure]

Figure 7. Factor profiles for natural gas, regional transport/photochemistry, and engine emissions factors.

[Figure]

Figure 7 (continue). Factor profiles for natural gas, regional transport/photochemistry, and engine emissions factors.

Line 323: This sounds more like a regional background. How do you know it's transport or active chemistry?

A: The high contribution of ozone makes this factor identifiable with photochemistry. And we are using "transport" to mean regional air masses transported to the measurement location, which essentially means the same thing as regional background.

Line 334: Toluene is also a known component of oil and gas extraction and is often in hydraulic

fracturing fluid.

A: Thank you for the critique, we have revised the text as follow: "Another important

emission source is oil and gas extraction (EPA, 1993)."

Reference: EPA, US Environmental Protection Agency, 1993, Locating and Estimating Sources of Toluene, https://www3.epa.gov/ttnchie1/le/toluene.pdf

References:

Demonstration of an Ethane Spectrometer for Methane Source Identification. T. I. Yacovitch, S. C. Herndon, J. R. Roscioli, C. Floerchinger, R. M. McGovern, M. Agnese, G. Petron, J. Kofler, C. Sweeney, A. Karion, S. A. Conley, E. A. Kort, L. Naehle, M. Fischer, L. Hildebrandt, J. Koeth, J. B. McManus, D. D. Nelson, M. S. Zahniser and C. E. Kolb. Environmental Science & Technology, 48(14), 8028-8034, doi:10.1021/es501475q, 2014

Source Signature of Volatile Organic Compounds from Oil and Natural Gas Operations in Northeastern Colorado. J. B. Gilman, B. M. Lerner, W. C. Kuster and J. A. de Gouw. Environmental Science & Technology, 47(3), 1297-1305, doi:10.1021/es304119a, 2013

Air Toxics and Other Volatile Organic Compound Emissions from Unconventional Oil and Gas Development. A. Hecobian, A. L. Clements, K. B. Shonkwiler, Y. Zhou, L. P. MacDonald, N. Hilliard, B. L. Wells, B. Bibeau, J. M. Ham, J. R. Pierce and J. L. Collett. Environmental Science & Technology Letters, 6(12), 720-726, doi:10.1021/acs.estlett.9b00591, 2019

**Comments from Reviewer #2**

Authors would like to thank reviewer #2 for the detailed review, critiques, and suggestions. Authors' responses are under each comment and a revised manuscript is provided with marked-up changes.

**Q1:** "The objective of this study is to investigate the effect of unconventional natural gas development activities on local air quality as observed at an active Marcellus Shale well pad" it would seem the objective is better stated as investigating the "emissions at the well pad", not the effect on local air quality, which is still interesting, but different. Please consider revising.
A: We regret for the mistake. We have revised the text as follow: "The objective of this study is to investigate the  emissions  at an active Marcellus Shale well pad at the Marcellus Shale Energy and Environment Laboratory (MSEEL) in Morgantown, Western Virginia, USA."

**Q2:** "Moreover, model results suggest that the major contributions to the regional transport/photochemistry factor occurred during horizontal drilling and drillout stages." This is just a relative shift where there are less emissions at the site so the background factor appears more pronounced. As it reads, one could think it was contributions to photochemistry or nearby photochemistry–worth clarifying.
A: We regret for the confusion. We have revised the text as follow: "Moreover, model results suggest that the  *regional transport/photochemistry* factor is more pronounced  during horizontal drilling and drillout due to limited emissions at the site. "

**Q3:** Fig 1: where is the monitoring lab on this map? Lines 259-262 discuss a lot of basic details about the figure that would be better in the caption, with the results and discussion section focused on substantive observations.

A: An ambient air monitoring laboratory (18' trailer with ambient air sampled from inlets on the trailer roof) was situated at the northeastern corner of the MSEEL well pad, which is marked with a star symbol on Figure 1. We have revised Figure 1 and merged with a second figure that shows the location of the trailer with respect to the location of the wells and the boundaries of the well pad. We have added the following explanation: "Figure 1 shows the location of the trailer with respect to the location of the wells and the boundaries of the well pad. The distance between the wells and the trailer is 90 m."

[Figure]

Figure 1. Location of the Marcellus Shale Energy and Environment Laboratory and the four production wells.

**Q4:** Fig 3 caption seems to end abruptly. Also, adding dates in the caption for each phase would be extremely helpful.

A: We regret for the mistake. We corrected the format of the Figure 3 caption. There is not enough space for the dates. If we add the dates, it would be difficult to read the figure. Instead, we shared the unconventional natural gas production process activity diagram with dates in the SI.

**Q5:** An opportunity is missed to comment on the production volumes of the well and how that relates to the observed "natural gas" factor emissions. There could also be more discussion of this factor. Efforts like this to quantify emissions would be useful, or at least to discuss quantitatively the role of each factor in the observed emissions (2 of the factors)–this only exists in fig 5.

A: The time series of the total gas production for the four wells is publically available on mseel.org website. We have added a new figure in SI. You can see the date production started, and the steady increase in cumulative gas produced below. We do not know how this would inform the interpretation of the natural gas factor.

[Figure]

Figure S3. The time series of the total gas production for the four wells (mseel.org).

**Q6:** Fig 5 could be much more clear. I suggest incorporating color.

A: We would like to thank for the suggestion. We have revised Figure 5 and presented the results in color.

Figure 6. The three-factor solution fingerprints for Drilling through Production Monitoring

[Figure]

Period, $F_{peak}=1$.

**Q7:** It is unclear how the wind direction is driving the PMF results and the observed average concentrations of pollutants at the site (Figs 3-4). The authors state that "With wind direction at this location most frequently from the southwest (Figure 2), this position optimized the occurrences of the laboratory being downwind of the well pad." But the wind is still only coming from that direction _23% of the time and there is no clear understanding of how that changes over the course of the study. Some effort needs to be taken to demonstrate that this is not biasing the concentrations or PMF results, perhaps by conducting the PMF analysis with the wind isolated to that direction, or quadrant. Otherwise the wind direction could be a driver of relative differences in the PMF factors. For example, it could play a role in observations like this "The skewness of the data for this phase indicates that the data may not be normally distributed." (line 268)

A: Please see our response to Reviewer #1/Q1.

**Q8:** The SI states there is a lot more VOC data, anything interesting in there? Are there indeed significant aromatics in the Marcellus natural gas emissions as the data would suggest or is this just from the engine exhaust factor? Fig 5 shows it all on the natural gas factor, but the text talks about toluene with the engine factor. This is unclear and needs to be cleared up.

A: Please see our response to Reviewer #1/Q2.

**Q9:** Can the VOC data be used to further substantiate the observed PMF factors (beyondwhat is done so far, which is useful)? This is a hint of this that appears at line 317. This seems like a big data set, but most of it is just left to the list in the SI without any data.

A: We have analyzed all VOCs before deciding the parameters for the PMF model. The most significant compounds are included in the study. The rest of the VOCs are listed in the SI but excluded in the model. Please see our response to Reviewer #1/Q2.

**Q10:** "Propane and isobutane had the second and third highest average concentrations, respectively, for each phase of development." (line 286), might be useful to clarify and show a figure in the SI for this since the sentence is not fully clear.

A: We regret for the mistake, mentioning propane and isobutene in the paragraph is misleading, we deleted the statement.

Supplement:
Generally, this needs to be cleaned up with more. Some of the figures are hard to read and the final section on uncertainty estimation is very challenging to follow given its structure.

A: We regret for the mistake and reformatted the document.

**Q11:** Figure S4-5 could be quite useful, even in the main text if the format showed the periods of the different activities with sufficient image clarity.

A: We moved Figure S4 to the main text as Figure 4 on page 14.

---

## Author Response (AR2)

15 February 2021

Dear Prof. Karl,

We are grateful to the editor and reviewers for their time and constructive comments on our manuscript. We have implemented their comments and suggestions and wish to submit a revised version of the manuscript for further consideration in the journal. Changes in the initial version of the manuscript are highlighted (with 'tracked changes') for added sentences or strikethrough for deleted sentences in the revised version. Below, we also provide a point-by-point response explaining how we have addressed each of the reviewers' comments. We look forward to the outcome of your assessment.

Yours sincerely,

On behalf of the co-authors

Nur H. Orak, Ph.D.

Assistant Professor
Department of Environmental Engineering
Marmara University
Istanbul, Turkey

**"Identifying and Quantifying Source Contributions of Air Quality Contaminants during Unconventional Shale Gas Extraction" by Nur H. Orak et al.**

*\* Comments from Referees are in black, authors' responses are in green, changes are in red color.*

**Comments from Reviewer #1**

The author's made improvements to the article but many of the reviewer questions remain unanswered/unclear. This is a valuable dataset but the analysis and manuscript could still use improvement in my opinion. I completely understand that there will be future papers coming from this unique dataset, but it would be very helpful to the readers to know what is slated for future work to help put this analysis into better context.

A: We would like to thank Reviewer #1 for the detailed review and constructive comments. Our responses are under each comment and a revised manuscript is provided with marked-up changes.

**Q1:** Line 99-102: Hecobian specifically compares the emissions of distinct operation phases of natural gas extraction. The analysis presented here would be greatly improved by doing a more direct comparison of the emissions at each stage of well production with Hecobian (e.g., for lines 276-281). Both datasets are incredibly rare and valuable, so it would be really helpful to see if they compare for gas fields in different regions of the U.S. and would further help to put Orak measurements into context.

A: Thank you for the suggestion we have provided a more detailed comparison with the results of Hecobian, 2019. We have revised the results as follow (No Markup-Line 296):
"Hecobian et al. (2019) investigated the emissions during different well pad development phases to analyze emission rates in the Denver-Julesburg and Piceance basins in Colorado, US. They observed that emission rates of benzene and most VOCs were highest during flowback for both basins, on the other hand, they had much lower emission rates from the production phase, which can be related to the differences in duration of each phase (days to weeks). Light alkanes and benzene concentrations were higher during hydraulic fracturing. It is difficult to directly compare the VOCs concentrations of the two studies, because the proposed study is based on continuous data during each phase while Hecobian et al. (2019) collected 374 measurements from five drilling, eight fracking, nine flowback, one liquids load out, and 11 production sites to analyze emission rates."

**Q2:** The addition of the trailer location to Figure 1 is very helpful. This combined with Figure 2 highlights the fact that the wind was rarely from the SE sector where the majority of the drilling equipment, and presumably the drilling activity, was occurring. It would be really helpful to show what the air composition was as a function of wind direction. Was methane, ethane, etc. higher when wind was coming from the SE sector? Does Factor 1 of the PMF results have the highest contribution to the ambient measurements when wind is from this direction? This was one of the unaddressed questions (Reviewer 1, Q1). You should have a timeseries of each factor of your final PMF model that can then be analyzed as a function of any other

variable that was NOT included in the PMF such as wind direction. If the Natural Gas factor is most prevalent in the SE sector, then it further adds confidence in the analysis.

A: Thank you for the suggestion. The trailer was situated at the northeastern corner of the MSEEL well pad (Figure 1) with wind direction at this location most frequently from the southwest (Figure 2). This position was optimized the occurrences of the laboratory being downwind of the well pad. We have prepared two new figures to show that SW dominated the higher overall concentrations (Figure S4) and added the following explanation on line 307 (No Markup):

"Figure S4 shows the dominant wind directions on overall concentrations, as well as giving information on the different concentration levels. Pollution roses show which wind directions contribute most to overall mean concentrations. For all air quality species, southwestern winds controlling the overall mean concentrations at the well pad. To explore the relationship between methane and ethane, we conditioned ethane by methane. Figure S5 indicates that higher ethane concentrations are associated with the SW and higher methane concentrations. The results also show that lower ethane and methane concentrations contributed from the east; the highest methane concentrations were obscured by a relatively high ethane background."

[Figure]

Figure S4. Methane, Ethane, NOx, NO₂, NO, and Ozone pollution roses

[Figure]

**Frequency of counts by wind direction (%)**

Figure S5. Ethane pollution rose conditioned by Methane concentration.

In terms of time series of each factor, as you can see below the trend of factor contribution does not provide any meaningful knowledge by itself, therefore, we think it would be confusing to share this figure with the audience. Instead, we prepared Figure S6 to show the contribution of wind direction to each PMF factor and added the following sentence on line 315 (No Markup) "The highest contribution to the factors were provided from the SW data (Figure S6)."

[Figure]

Factor 1- Engine Emissions Factor

Factor 2- Natural Gas Factor

Factor 3- Regional Transport/Photochemistry Factor

[Figure]

[Figure]

Figure S6. The PMF factor contribution roses for Engine Emissions factor, Regional Transport/ Photochemistry factor, and Natural Gas factor.

**Q3:** Figure 3 is the key figure in this manuscript, in my opinion. This clearly shows the different chemical composition of the air measured during each of the important drilling/hydraulic fracturing/production steps. Why doesn't PMF pick up these differences and lump all into a single generic factor? How do these results compare to other studies?

A: Thank you for pointing out your concerns. We think Figure 6 is the main figure that serves to the objective of this study. As we mentioned in our first response, the research team is conducting several analysis that has different objectives and methods. We examined several factors to capture the optimal number and analyzed different fPeak values to explore the robustness of the selected PMF solution. We explained the rotational ambiguity of the factors to justify the selection. However, there are several limitations of PMF model. To be able to answer your questions, we have added the following explanation (No Markup-Line 370):

" PMF models have several limitations. First, it needs large datasets. In this study, the number of data varies based on the duration of the activity (Figure S2). Therefore, the contribution to the factors is not same for each phase. This is the main reason behind the uncertainty of defined factors. Second, the accuracy and precision of measured species limit the analysis. The determination of the number and character of factors is based on an expert's interpretation. Comprehensive information is needed on source profiles to verify the defined source profiles. Finally, the pre-set parameters are playing an important role on the model results."

**Q4:** Lines 288-292: Be sure to specify the units as you are comparing the ethane (ppb) to methane (ppm) ratioor else add 10^-3 to the ratios. How does the ethane to methane ratio compare to other ONG emissions (i.e., Yakovitch et al. and may other papers)?

A: We regret for the mistake, we have added the unit on Figure 4. For the comparison please see our response to Q1.

**Q5:** Lines 328: Source Profiles. Another useful reference regarding PMF analysis in an oil and gas field to add to the discussion could be:
Source characterization of volatile organic compounds in the Colorado Northern Front Range Metropolitan Area during spring and summer 2015. A. Abeleira, I. B. Pollack, B. Sive, Y. Zhou, E. V. Fischer and D. K. Farmer. 122(6), 3595-3613, doi:https://doi.org/10.1002/2016JD026227, 2017

A: Thank you for the suggestion. We had added the suggested reference.

**Q6:** Figure 6: I'm still trying to figure out why CO2 and methane have virtually the same PMF factor fingerprints. Clearly, the natural gas factor isn't just natural gas as raw natural gas does NOT contain NOx and I would not expect it to be composed primarily of CO2. Also, the engine emissions factor doesn't contain an appreciable amount of CO2. Why? It seems to me that the PMF factors aren't fully resolving in a meaningful way. Also, why do hexane and benzene not have any attribution to "regional transport" as these two species are sufficiently long-lived in the atmosphere to have a significant background, much more so than toluene that has ~25% attributed to transport?

A: We do understand your concerns. There are several limitations of the study. Please see our response to Q3. PMF has limitations and the factors are not usually perfectly resolved. The signal for each measured species can have something to do with it, too. So although they are valid data points, they maybe do not get resolved into factors as well as if the signal was stronger.

**Q7:** The last, "big picture", piece of this analysis that is missing is how the natural gas drilling activities actually affected air quality, which is stated as being the motivation for this paper. Since you have the PMF factors for each species, then you should be able to answer the question of how the air quality would be different if the "natural gas" factor was removed or how it compared to the other factors by calculating an Air Quality Index or OH-reactivity, or some other metric.

A: Thank you for the suggestion. It would be a separate scenario to analyze the impacts on air quality if we remove the natural gas factor. We think the proposed analysis is beyond the scope of this study. We have explained the big picture on line 394 (No Markup):

"As determined by the PMF analysis, a measurable increase in natural gas-related pollutant concentrations and the associated natural gas factor contribution from different stages of active phase was not observed. At the downwind distance of 600m from the well pad center to the air monitoring laboratory, the emissions from the well pad were not easily distinguishable from typical variations in ambient background concentrations. West Virginia has many natural gas wells that contribute to the ambient background, as evidenced by ethane concentrations that are higher than typical global background (Rinsland et al. 1987; Rudolph et al. 1996). Short-lived peak events that were observed when the wind direction was coming from the well pad show that emissions can be dispersed downwind and detected at this distance, but when concentrations are averaged and analyzed with a PMF analysis the peak events were not significant enough to result in a measurable impact of the well pad emissions at the receptor location. Understanding the air quality impacts of operational phases is important since it has potential to help inform future decision-making and constrain cumulative impact assessments."